# The Clinical Characteristics of Bloodstream Infections Due to *Candida* spp. in Patients Hospitalized in Intensive Care Units during the SARS-CoV-2 Pandemic: The Results of a Multicenter Study

**DOI:** 10.3390/jof9060642

**Published:** 2023-06-01

**Authors:** Francesco Pallotta, Lucia Brescini, Arianna Ianovitz, Ilenia Luchetti, Lucia Franca, Benedetta Canovari, Elisabetta Cerutti, Francesco Barchiesi

**Affiliations:** 1Dipartimento di Scienze Biomediche e Sanità Pubblica, Università Politecnica delle Marche, 60126 Ancona, Italy; pallottafrancesco1993@gmail.com (F.P.); arianna.ianovitz@gmail.com (A.I.); ilenialuchetti@hotmail.it (I.L.); lucia_franca@libero.it (L.F.); f.barchiesi@staff.univpm.it (F.B.); 2Clinica Malattie Infettive, Azienda Ospedaliera Universitaria Ospedali Riuniti Umberto I-Lancisi-Salesi, 60126 Ancona, Italy; 3Malattie Infettive, Azienda Sanitaria Territoriale Pesaro-Urbino, 61029 Pesaro, Italy; benedetta.canovari@ospedalimarchenord.it; 4Anestesia e Rianimazione dei Trapianti e Chirurgia Maggiore, Azienda Ospedaliera Universitaria Ospedali Riuniti Umberto I-Lancisi-Salesi, 60126 Ancona, Italy; elisabetta.cerutti@ospedaliriuniti.marche.it

**Keywords:** COVID-19, candidemia, SARS-CoV-2, ICU

## Abstract

Candidemia is a serious health threat. Whether this infection has a greater incidence and a higher mortality rate in patients with COVID-19 is still debated. In this multicenter, retrospective, observational study, we aimed to identify the clinical characteristics associated with the 30-day mortality in critically ill patients with candidemia and to define the differences in candidemic patients with and without COVID-19. Over a three-year period (2019–2021), we identified 53 critically ill patients with candidemia, 18 of whom (34%) had COVID-19 and were hospitalized in four ICUs. The most frequent comorbidities were cardiovascular (42%), neurological (17%), chronic pulmonary diseases, chronic kidney failure, and solid tumors (13% each). A significantly higher proportion of COVID-19 patients had pneumonia, ARDS, septic shock, and were undergoing an ECMO procedure. On the contrary, non-COVID-19 patients had undergone previous surgeries and had used TPN more frequently. The mortality rate in the overall population was 43%: 39% and 46% in the COVID-19 and non-COVID-19 patients, respectively. The independent risk factors associated with a higher mortality were CVVH (HR 29.08 [CI 95% 3.37–250]) and a Charlson’s score of > 3 (HR 9.346 [CI 95% 1.054–82.861]). In conclusion, we demonstrated that candidemia still has a high mortality rate in patients admitted to ICUs, irrespective of infection due to SARS-CoV-2.

## 1. Introduction

*Candida* spp. is a frequent agent of infection in hospitalized patients and is responsible for high mortality, especially in patients admitted to ICUs (intensive care unit) [1,2,3]. In a European multicenter study [4], it was calculated that the incidence of invasive candidiasis in ICUs was 7.07 per 1000 ICU admissions. Candidemia is the most frequent form of invasive candidiasis observed, representing the seventh leading cause of nosocomial bloodstream infections in Europe [5].

Some studies have noted an increasing trend in the incidence of candidemia in ICUs [2,6], which can be explained by the risk factors for critically ill patients, such as the presence of central venous catheters (CVC), previous exposure to broad-spectrum antibiotic therapies, invasive procedures, and the length of the ICU stay. Although *Candida albicans* is the most frequently isolated species, a shift toward non-*albicans* species, with some of them presenting multiple antifungal resistance patterns, represents a worrisome trend [7,8].

In recent years, the SARS-CoV-2 pandemic has been responsible for a great number of ICU admissions of critically ill patients with acute respiratory distress syndrome (ARDS), with many requiring invasive mechanical ventilation, high-dose corticosteroids, and extracorporeal membrane oxygenation [9,10,11,12]. A greater incidence of candidemia and a higher mortality is seen in these critically ill Coronavirus disease 2019 (COVID-19) patients [13,14,15,16].

The primary aim of this study was to describe the clinical characteristics of bloodstream infections due to *Candida* spp. in patients hospitalized in intensive care units during the SARS-CoV-2 pandemic. The secondary aim was to define the differences in terms of the clinical and prognostic risk factors between COVID-19 and non-COVID-19 patients.

## 2. Patients and Methods

### 2.1. Hospital Setting, Study Design, Data Collection, and Definitions 

This retrospective multicenter study was conducted in three hospitals in the Marche region of Italy. The hospitals involved were: Azienda Ospedaliero Universitaria delle Marche, a tertiary referral hospital located in Ancona, and Azienda Ospedaliera Ospedali Riunti Marche Nord, which consists of two secondary hospitals (Pesaro and Fano hospitals). There were two ICUs that admitted critically ill COVID-19 patients and one ICU each in the Pesaro and Fano hospitals. All the cases of candidemia occurred in adult patients (>18 years old) hospitalized in these ICUs between 1 January 2019 and 31 December 2021, and they were retrieved from the microbiology laboratory database. Data regarding demographic characteristics and clinical risk factors were collected from the patients’ medical records. The Institutional Review Board of the “Azienda Ospedaliero-Universitaria Ospedali Riuniti Umberto I-Lancisi-Salesi” (Coordinator center) granted retrospective access to these data without the need for individual informed consent. This consent was not sought since the data were analyzed anonymously.

A case of *Candida* bloodstream infection (BSI) was defined as a peripheral isolation of *Candida* spp. from a blood culture in a patient with temporally related clinical signs and symptoms of infection. Appropriate antifungal therapy was considered when an appropriate drug (based on subsequent in vitro susceptibility testing results) with an adequate dosage was started within 72 h from the first blood culture performed. An adequate dosage of an antifungal agent was defined according to the IDSA 2009–2016 guidelines [17,18]. Early central venous catheter (CVC) removal was defined a removal of the line within 48 h from drawing the blood culture. The mortality was calculated after 30 days from the occurrence of the episode of *Candida* BSI.

### 2.2. Microbiological Methods

*Candida* species were isolated from blood samples using BacT/ALERT (bioMe’rieux) and identified with the MALDI-TOF Biotyper (Bruker Daltonics, Bremen, Germany). Antifungal susceptibility testing was performed for fluconazole, caspofungin, and amphotericin B using the SensititreYeastOne colorimetric plate (Trek Diagnostic System). The MIC results were interpreted according to the latest species-specific clinical breakpoints, as established by the Clinical and Laboratory Standards Institute (CLSI) [19]. 

All patients with COVID-19 had a SARS-CoV-2 infection confirmed using a reverse-transcriptase polymerase chain reaction assay.

### 2.3. Statistical Analysis

Categorical variables were expressed as absolute numbers and their relative frequencies; continuous variables were expressed as median and interquartile ranges (IQR). The categorical variables were compared using the χ^2^ or Fisher exact test, while the continuous variables were evaluated using the Student *t*-test (for normally distributed variables) or Mann–Whitney U test (for non-normally distributed variables). The variables that reached a statistical significance (*p* < 0.05) in the univariate analysis were analyzed using a multivariate logistic regression analysis to identify the independent risk factors for 30-day mortality. The results were expressed as odds ratios and 95% CIs. All the statistical analyses were performed using the statistical package SPSS for Windows v. 20 (SPSS Inc., Chicago, IL, USA). A *p* value of <0.05 was considered to represent statistical significance and all the statistical tests were two-tailed.

## 3. Results

### 3.1. Characteristics of Patients

During the study period, 53 cases of candidemia were diagnosed in four ICUs. Eighteen cases (34%) occurred in COVID-19 patients. The baseline characteristics of the study population are reported in Table 1. The majority were male (71%), with a median age of 67.5 years. Cardiovascular disease was the most frequent comorbidity (42%) and 42% of the patients had a Charlsons CI of ≥ 3. CVC was present in all the patients.

The most common acute complications during candidemia were pneumonia (64%) and ARDS (49%). Septic shock and acute kidney failure occurred in 32% and 41% of cases, respectively. Most patients (87%) were intubated and placed on mechanical ventilation. In total, 36% of patients were placed on haemodialysis and 19% in veno-venous extracorporeal membrane oxygenation (ECMO). Previous antibiotic therapy had been given in all cases, while 28% of the patients had received previous antifungal therapy before the onset of candidemia.

### 3.2. Mycological Tests Results and Treatments

*Candida albicans* was the species most frequently isolated (50%), followed by *C. parapsilosis* (28%), *C. glabrata* (11%), *C. tropicalis* (2%), and other *Candida* spp. (8%). The latter group included one isolate each of *M. guilliermondii, P. krusei, C. lusitaniae*, and *C. dubliniensis.* All the *Candida* isolates were susceptible to azoles (except for the one isolate of *P.kudriavzevii*, which is intrinsically resistant to azoles, and the one isolate of *C. glabrata*, which is susceptible and dose-dependent, according to CLSI), echinocandins, and amphotericin B. Primary antifungal treatment was considered appropriate in 66% of cases: echinocandins were the most frequently prescribed drugs (49%), followed by azoles (25%). 

### 3.3. Outcome of Patients with Candidemia and 30-Day Mortality Risk Factors

The crude mortality on day 30 was 43% in the overall population: 39% (7/18) and 46% (16/35) in the COVID-19 and non-COVID-19 patients, respectively (*p* = 0.225). The following characteristics were significantly more common in patients with a negative outcome: older age, the presence of cardiovascular diseases, a higher Charlson’s score, being on haemodialysis, and the presence of septic shock (*p* ranging from <0.0001 to 0.023). Contrarily, being on mechanical ventilation in the 72 h before the onset of candidemia was more frequent in the surviving patients (*p* 0.034). A multivariate analysis showed that only haemodialysis (HR 24,840 [CI 95% 3.35–184.00], *p* = 0.002) and a Charlson’s CI of > 3 (HR 9.346 [CI 95% 1.054–82.861], *p* = 0.045) were independent risk factors for a higher mortality (Table 2).

### 3.4. Differences between COVID+ and COVID—Patients with Candidemia

The differences between the COVID-positive and -negative patients are reported in Table 3. The following characteristics were significantly more common in the COVID-19 patients: being on ECMO, suffering from ARDS, pneumonia, and septic shock (*p* ranging from <0.0001 to 0.003). On the other hand, previous surgery and being on total parenteral nutrition (TPN) were more common in the non-COVID patients (*p* < 0.0001 and 0.018). 

## 4. Discussion

In this study, we analyzed the characteristics of candidemia in patients admitted to the ICUs of three hospitals in our region during the SARS-CoV-2 pandemic.

Several studies described an increased incidence of candidemia in patients admitted to an ICU during the SARS-CoV-2 pandemic, especially in COVID-19 patients [7,20,21,22,23]. This is in contrast with the study of Blaize et al. [24], who identified only 13 cases of candidemia in an 11-month observation period, demonstrating a low incidence of candidemia in critically ill COVID-19 patients. Similarly, although we did not investigate the incidence of candidemia, we identified a greater number of cases of *Candida* BSI in patients admitted to ICUs for reasons other than COVID-19. In a previous study conducted in our region during the first wave of the pandemic, we compared the BSIs of COVID and non-COVID patients hospitalized either in critical or non-critical care units and found that the incidence of BSI caused by *Candida* spp. was greater in the COVID-19 patients [25]. The differences in the study population (i.e., ICUs vs. all hospital units) and the time period of the pandemic (i.e., first vs. the following waves) might explain this discrepancy. 

BSIs caused by *Candida* spp. are very serious forms of infection, burdened by a particularly high mortality. This is especially true for patients admitted to ICUs who are frequently clinically unstable because of acute complications, often presenting with underlying chronic comorbidities. These characteristics explain the worse outcomes that are frequently observed for critically ill patients with candidemia. In our patient population, we found a 30-day crude mortality rate of 43%, which is in line with that of other studies. [4,20].

The factors that were identified as being associated with a worse outcome were having a Charlson’s comorbidity index of ≥3 and being on haemodialysis. 

A greater Charlson’s score reflects a greater burden of comorbidities and a lesser 10-year survival rate. Several studies have shown that the Charlson’s comorbidity index is related to the mortality in patients with candidemia [26,27]. Although in the present study we did not identify any particular comorbidity as being an independent risk factor for this mortality, previous literature data have described solid tumors, chronic kidney disease, and liver cirrhosis as predisposing factors for a negative outcome [28,29,30]. 

Haemodialysis is a well-recognized risk factor for the development of candidemia [31]. Patients who require haemodialysis need the insertion of a CVC, which can be colonized by bacteria and yeast of the skin microbiota, thereby representing an entry site for bloodstream infections. Secondarily, acute kidney injuries requiring haemodialysis often occur in critically ill patients, especially in septic shock patients. For this reason, patients on haemodialysis are generally more seriously ill, explaining its association with an increased mortality in the setting of candidemia.

The most widespread species in our institution was *Candida albicans*, which is similar to studies conducted in other countries [4,29,32,33]. Non-*albicans Candida* species accounted for 53% of all isolates, with *Candida parapsilosis* being the most common. The diffusion of antifungal resistance represents a major concern in ICUs, especially among non-*albicans Candida* species, which are more often becoming responsible for candidemia in critically ill patients. For example, in the study of Ramoz-Martìnez et al. [7], 15 of their 20 fluconazole-resistant isolates were *C. parapsilosis*, while in Alameda country [28], the dominant species implicated in candidemia was *C. glabrata*. Another alarming issue is the diffusion of *C. auris*, which frequently shows a pan-antifungal resistant phenotype and has been isolated in a great number of countries around the world, even in Italy [8,34,35]. 

Although in several studies, candidemia proved to increase the risk of death in patients with COVID-19 [15,36,37], we found that the 30-day mortality in our population was not influenced by SARS-CoV-2 infection status. Rather, for the COVID-19 patients, we found a mortality rate quite lower than that observed for the general population (39% (7/18) vs. 46% (19/35), respectively). This can be explained by the low number of candidemia cases identified in critically ill patients. 

The COVID-19 patients more frequently presented with ARDS, pneumonia, and septic shock and were more frequently on ECMO. This is in line with the clinical presentation of severe SARS-CoV-2 infection, which frequently manifests with lung involvement that requires mechanical ventilation because of a decrease in the PaO_2_/FiO_2_ ratio, with a necessity for ECMO use in some cases [38]. On the contrary, the non-COVID-19 patients had undergone previous surgeries and used TPN more frequently. Both these features are reported to be classical risk factors for the development of invasive candidiasis [39,40]. 

Despite our efforts, our study has some limitations, which are primarily linked to its retrospective nature and the limited number of identified cases. Furthermore, the lack of a control group of patients without candidemia precludes any inferences about the risk factors for candidemia in the setting of severe COVID-19. 

In conclusion, candidemia remains a serious health threat to critically ill patients, with an overall mortality of 43% in our experience. Although we did not demonstrate a correlation between SARS-CoV-2 infection and mortality in the setting of candidemia, further studies are warranted to identify the possible modifiable risk factors in this group of patients. 

## Figures and Tables

**Table 1 jof-09-00642-t001:** Characteristics of patients with candidemia hospitalized in ICU.

Characteristics	All Patients (*n* = 53)	Not Surviving *n* = 23 (43%)	Surviving *n* = 30 (57%)	*p* Value ^a^
Male sex, *n* (%)	37 (71%)	17 (77%)	20 (67%)	0.600
Age, median (IQR) ^b^	67.5 (54–75)	73 (62.5–76.5)	56.5 (53–75)	0.003
Chronic comorbidities				
Chronic pulmonary diseases, *n* (%) ^c^	7 (13%)	4 (17%)	3 (10%)	0.705
Hematological malignancy, *n* (%)	2 (4%)	1 (4%)	1 (3%)	1
Cardiovascular diseases, *n* (%) ^d^	22 (42%)	14 (61%)	8 (27%)	0.023
Neurological diseases, *n* (%) ^e^	9 (17%)	4 (17%)	5 (17%)	1
Gastrointestinal diseases, *n* (%) ^f^	6 (11%)	3 (13%)	3 (10%)	1
Diabetes mellitus, *n* (%)	4 (9%)	1 (6%)	3 (12%)	0.634
Chronic renal failure, *n* (%)	7 (13%)	4 (17%)	3 (10%)	0.451
Solid tumors, *n* (%)	7 (13%)	4 (17%)	3 (10%)	0.451
Solid organ transplant, *n* (%)	3 (6%)	1 (4%)	2 (7%)	1
Chronic hepatitis, *n* (%)	4 (8%)	0	4 (13%)	0.124
Surgery, *n* (%)	23 (43%)	11 (48%)	12 (40%)	0.772
Charlson’s score > 3	22 (42%)	19 (83%)	12 (40%)	0.005
Central venous catheter, *n* (%)	53 (100%)	23 (100%)	30 (100%)	
Early central venous catheter removal, *n* (%) ^g^	21 (40%)	12 (52%)	9 (31%)	0.208
Previous invasive procedures (<72 h), *n* (%) ^h^	10 (19%)	6 (26%)	4 (13%)	0.300
Mechanical ventilation, *n* (%)	46 (87%)	17 (74%)	29 (97%)	0.034
Parenteral nutrition, *n* (%)	31 (59%)	17 (74%)	14 (47%)	0.087
Haemodialysis, *n* (%)	19 (36%)	15 (65%)	4 (13%)	<0.0001
ECMO, *n* (%)	10 (19%)	3 (13%)	7 (24%)	0.482
ARDS, *n* (%)	26 (49%)	11 (48%)	15 (50%)	1
Steroid therapy, *n* (%)	36 (68%)	17 (74%)	19 (63%)	0.602
Immunosuppressive therapy, *n* (%) ^i^	8 (15%)	5 (21%)	3 (10%)	0.272
Neutropenia, *n* (%)	2 (4%)	2 (9%)	0	0.184
Pneumonia, *n* (%)	34 (64%)	15 (65%)	19 (63%)	1
Septic shock, *n* (%)	17 (32%)	12 (52%)	5 (17%)	0.014
Acute kidney failure, *n* (%)	22 (41%)	12 (52%)	10 (33%)	0.272
Bleeding, *n* (%)	20 (38%)	11 (48%)	9 (30%)	0.298
Concomitant bacteriemia, *n* (%)	38 (72%)	14 (71%)	24 (80%)	0.221
Pre-infection hospitalization, median days (IQR)	15 (5.25–32.25)	14,5 (5–30)	15 (6–33)	0.638
Previous antibiotic therapy, *n* (%)	52 (98%)	22 (96%)	30 (100%)	0.434
Previous antifungal therapy, *n* (%)	15 (28%)	7 (30%)	8 (27%)	1
*Candida* species				
*Candida albicans*, *n* (%)	27 (50%)	11 (48%)	16 (53%)	0.438
*Candida parapsilosis*, *n* (%)	15 (28%)	5 (22%)	10 (33%)	
*Candida tropicalis*, *n* (%)	1 (2%)	1 (4%)	0	
*Candida glabrata*, *n* (%)	6 (11%)	4 (17%)	2 (7%)	
Other *Candida* species, *n* (%) ^j^	4 (8%)	2 (9%)	2 (7%)	
Appropriate antifungal therapy, *n* (%) ^k^	35 (66%)	14 (61%)	21 (70%)	0.687
Primary antifungal therapy				
Azoles, *n* (%)	13 (25%)	4 (18%)	9 (30%)	0.114
Echinocandins, *n* (%)	26 (49%)	11 (48%)	15 (50%)	0.546
Polyenes, *n* (%)	1 (2%)	1 (4%)	0	0.409
No treatment, *n* (%)	12 (23%)	7 (30%)	5 (17%)	0.429
COVID disease, *n* (%)	18 (34%)	7 (30%)	11 (37%)	0.855

^a^ Categorical variables were compared using the χ^2^ or Fisher exact test, while continuous variables were evaluated using the Student *t*-test or the Mann–Whitney *U* test. ^b^ IQR, Interquartile range. ^c^ Chronic pulmonary diseases include asthma, chronic bronchitis, emphysema, and lung fibrosis. ^d^ Cardiovascular diseases include heart failure, ischemic heart disease, endocarditis, and arrhythmia. ^e^ Neurological diseases include Parkinson’s disease, Alzheimer’s disease, and paralysis. ^f^ Gastrointestinal diseases include Crohn’s disease, ulcerative colitis, chronic pancreatitis, and gallbladder stones. ^g^ Early central venous catheter removal was considered occurring within 48 h from blood cultures drawing. ^h^ Previous invasive procedures include endoscopy and positionning of any device. ^i^ Immunosuppressive therapy includes calcineurin inhibitors and monoclonal antibodies. ^j^ Other *Candida* species include *Meyerozyma guilliermondii* (*n* = 1), *Clavispora lusitaniae* (*n* = 1), *Candida dubliniensis* (*n* = 1), and *Pichia kudriavzevii* (*n* = 1). ^k^ Antifungal therapy was considered appropriate when the appropriate drug with adequate dosage was started within 72 h of the first blood culture being performed.

**Table 2 jof-09-00642-t002:** Multivariate analysis of risk factors for 30-day mortality in the study cohort.

Risk Factors	Hazard Ratio	CI 95%		*p* Value
		Lower Limit	Upper Limit	
Haemodialysis	29.080	3.37	250	0.02
Charlson’s score > 3	9.346	1.054	82.861	0.045

**Table 3 jof-09-00642-t003:** Univariate analysis of patients with Candidemia with and without COVID-19.

Characteristics	Non COVID-19 *n* = 27	COVID-19 *n* = 17	*p* Value
Surgery, *n* (%)	22 (63%)	1 (6%)	<0.001
Parenteral nutrition, *n* (%)	25 (71%)	6 (33%)	0.018
ECMO, *n* (%)	2 (6%)	8 (44%)	0.002
ARDS, *n* (%)	8 (23%)	18 (100%)	<0.00.1
Pneumonia, *n* (%)	16 (46%)	18 (100%)	<0.00.1
Septic shock, *n* (%)	0	11 (41%)	0.003

## Data Availability

The data are unavaible due to privacy restrictions in public datasets, but they can be requested from the corresponding author, who is in possession of them.

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
