# Peer review of "The Clinical Characteristics of Bloodstream Infections Due to Candida spp. in Patients Hospitalized in Intensive Care Units during the SARS-CoV-2 Pandemic: The Results of a Multicenter Study"

_jof, 2023, doi:10.3390/jof9060642_

Round 1

Reviewer 1 Report

The manuscript " Clinical characteristics of bloodstream infections....." discusses a retrospective observational study of candidemia in  COVID-19 and non-COVID-19 patients with risk factors. The study is short with a low number of patients but is well conducted and the result is well written. statistical analysis was performed as required. The following minor comments need to be considered to improve the readability of the manuscript. 

1- Always the full name should be written when the abbreviated terms are mentioned for the first time in the manuscript. for example ( CVVH, ECMO....).

2- Line 78-79: The sensititre yeastOne panel contains about 8 antifungal agents. If the author used this antifungal susceptibility testing, why only 3 antifungal agents were considered for this study? 

3- Line 83: Which method was used for the molecular assay of COVID-19: Author can you give a short detail on the method and reference if available?

4-In the legend of table-1 and in the discussion section: other uncommon and rare Candida species isolated were mentioned in the Anamorpth state and it would be currently, more acceptable to use the Telemorph nomenclature. Author, please correct the scientific names to read as follows

Candida guillermondiiMeyerozyma  guillermondii

Candida lusitaniae    =    Clavispora lusitaniae

Candid krusei             =    Pichia kudriavzevii 

Reviewer 2 Report

Dear authors:

This is an interesting study about candidemia in ICU’s patients.

I found a few mistakes:

1.       spp. (species) should not be written in italics.

2.       Please explain all acronyms you use the first time they appear in the text.

3.       You mention that all the Candida isolates were susceptible to the azoles, echinocandins and amphotericin B. But you isolated one C. krusei that is intrinsically resistant to fluconazole, also C. glabrata only could be considered susceptible dose dependent but not susceptible, according to Document M60 from CLSI.  There are only ECV for other Candida spp. Please explain.

4.       In table 1 it appears that there are differences statistically significant in median age, but the median age of both groups was the same (with a different IQR).

5.       Mechanical ventilation was used in 97% of patients who survived, but you say it was one of the factors of negative outcome (line 138).

6.       In the discussion you mention as a limitation of this study (lines 211-213), the lack of a control group of patients without candidemia, but I believe in the ICU of those hospitals you must have cared patients with COVID-19 infection and without candidemia who could have been included in that control group during the period analyzed.

Reviewer 3 Report

The aim of this epidemiological study is very interesting, especially at this moment when the COVID pandemic is not being considered as the main health problem in ICU units. Mortality due to candidemia is indeed a huge problem. Overall, the paper is well written and clear.

Nevertheless, I think that, given that the authors had access to patients' clinical charts, more interesting information could be analyzed. On the other hand, some issues should be clarified. 

1) Please, don't use abbreviations unless you have previously described what they stand for.

2) A major remark is that the study period begins a year before the COVID epidemic in Italy. Please explain if this was done to enroll more patients or for another reason.

3) Lines 62-66: Is this in accordance with national regimentation on clinical investigation? Why was the study submitted to ethical consideration at a non participating hospital?

4) Lines 97-109: It would be very interesting to know what was the cause of admission to ICU. The analysis could give different results if causes of admission were considered as COVID or non-COVID, for instance. It would also be interesting to know the source of candidemia whenever possible.

5) Table 1: How is it possible that survivors and non-survivors mean age is significantly different? For the purposes of this study, comparison of death rates only have sense if the cause of death is known. How many patients died from COVID? How many from candidemia? How many for other causes? Related to point 1: how many patients died before and after 2020?

6) Lines 158-159: in the discussion it is stated that more patients with candidemia were admitted to ICU for reasons other than COVID. However, this is not described in the results.
